# Rest-Katyusha: Exploiting the Solution's Structure via Scheduled Restart Schemes

**Junqi Tang**
School of Engineering
University of Edinburgh, UK
J.Tang@ed.ac.uk

**Mohammad Golbabaee**
Department of Computer Science
University of Bath, UK
M.Golbabaee@bath.ac.uk

**Francis Bach**
INRIA - ENS
PSL Research University, France
Francis.Bach@inria.fr

**Mike Davies**
School of Engineering
University of Edinburgh, UK
Mike.Davies@ed.ac.uk

## Abstract

We propose a structure-adaptive variant of a state-of-the-art stochastic variance-reduced gradient algorithm Katyusha for regularized empirical risk minimization. The proposed method is able to exploit the intrinsic low-dimensional structure of the solution, such as sparsity or low rank which is enforced by a non-smooth regularization, to achieve even faster convergence rate. This provable algorithmic improvement is done by restarting the Katyusha algorithm according to restricted strong-convexity (RSC) constants. We also propose an adaptive-restart variant which is able to estimate the RSC on the fly and adjust the restart period automatically. We demonstrate the effectiveness of our approach via numerical experiments.

## 1   Introduction

Many applications in supervised machine learning and signal processing share the same goal, which is to estimate the minimizer of a population risk via minimizing the empirical risk $\frac{1}{n}\sum_{i=1}^{n} f_i(a_i, x)$, where $a_i, x \in \mathbb{R}^d$, each $f_i$ is a convex and smooth function [1]. In supervised machine learning, $a_i$ is often referred to as the training data sample, while in signal/image processing applications it is the representation of measurements. In practice the number of data samples or measurements is limited, and from them we attempt to infer $x^\dagger \in \mathbb{R}^d$ which is the unique minimizer of the population risk:

$$x^\dagger = \arg \min_x \mathbb{E}_a \bar{f}(a, x). \tag{1}$$

The ultimate goal is to get a vector $x^\star$ which is a good approximation of $x^\dagger$ from the empirical risk. Since in many interesting applications, the dimension of parameter space $d$ is of the same order or even larger than the number of data samples $n$, minimizing the empirical risk alone will introduce overfitting and hence leads to poor estimation of the true parameter $x^\dagger$ [2, 3]. In general, avoiding overfitting is a key issue in both machine learning and signal processing, and the most common approach is to add some regularization while minimizing the empirical risk [4, 5, 6]:

$$x^\star \in \arg \min_{x \in \mathbb{R}^d} \left\{ F(x) := f(x) + \lambda g(x) \right\}, \quad f(x) := \frac{1}{n} \sum_{i=1}^{n} f_i(x), \tag{2}$$

where for the sake of compactness of notation, we denote $f_i(x) := f_i(a_i, x)$. Each $f_i$ is assumed to be convex and have $L$-Lipschitz continuous gradient, while the regularization term $g(x)$ is assumed to be a simple convex function and possibly non-smooth.

## 1.1 Accelerated Stochastic Variance-Reduced Optimization

To handle the empirical risk minimization in the "big data" and "big dimension" regimes, stochastic gradient-based iterative algorithms are most often considered. The most basic one is often referred to as *stochastic gradient descent* (SGD) [7, 8], in every iteration of which only one or a few functions $f_i$ are randomly selected, and only their gradients are calculated as an estimation of the full gradient. However, the convergence rate of SGD is sub-linear even when the loss function $F$ is strongly-convex.

To further accelerate the stochastic gradient descent algorithm, researchers have recently developed techniques which progressively reduce the variance of stochastic gradient estimators, starting from SAG [9, 10], SDCA [11], then SVRG [12, 13] and SAGA [14]. Such methods enjoy a linear convergence rate when the cost function $F$ is $\mu$-strongly-convex and each $f_i$ has $L$-Lipschitz continuous gradients, that is, to achieve an output $\hat{x}$ which satisfies $F(\hat{x}) - F(x^\star) \leq \delta$, the total number of stochastic gradient evaluations needed is $O(n + L/\mu) \log \frac{1}{\delta}$. *Nesterov's acceleration* [15, 16, 17] has also been successfully applied to construct variance-reduced methods which have an accelerated linear-convergence rate [18, 19, 20, 21, 22, 23, 24, 25]:

$$O\left(n + \sqrt{\frac{nL}{\mu}}\right) \log \frac{1}{\delta}. \tag{3}$$

It is worth noting that this convergence rate has been shown to be worst-case optimal [21]. However, all of these algorithms need explicit knowledge of the strong-convexity parameter $\mu$. Very recently, [26] has shown theoretically that it is *impossible* for an accelerated incremental gradient method to achieve this ideal linear rate without the knowledge of $\mu$. Since in general the strong-convexity is hard to be estimated accurately, researchers propose adaptive restart schemes [27, 28, 29, 30, 25, 31] for accelerated first-order methods, either by the means of enforcing monotonicity on functional decay, or by estimating the strong-convexity on the fly.

## 1.2 Solution's Structure, Restricted Strong Convexity, and Faster Convergence

In many interesting large-scale optimization problems in machine learning, the solution $x^\star$ in (2) has some low-dimensional structure such as sparsity [4], group-sparsity [32], low-rank [33] or piece-wise smoothness [6], enforced by the non-smooth regularization. It is intuitive that an optimal algorithm for this type of problem should take into account and exploit such solution's structure. We believe that, when being utilized properly, this prior information of the solution will facilitate the convergence of an iterative algorithm.

One important theoretical cornerstone is the *modified restricted strong convexity* framework presented by Agarwal et al. [34]. In the context of statistical estimation with high-dimensional data where the usual strong-convexity assumption is vacuous, these authors have shown that the proximal gradient descent method is able to achieve global linear convergence up to a point $x$ which satisfies $\|x - x^\star\|_2 = o(\|x^\star - x^\dagger\|_2)$, the accuracy level of *statistical precision*. Moreover, the results based on this restricted strong-convexity framework indicate that the convergence rate of the proximal gradient become faster when the model complexity of the solution is lower.

Inspired by Agarwal et al. [34], Qu and Xu [35] extend this framework to analyse some variance-reduced stochastic gradient methods such as proximal SVRG [13]. Most recently, based on the same framework, researchers proposed a two-stage APCG algorithm [36, 37], an accelerated coordinate descent method able to exploit the solution's structure for faster convergence. Moreover, in the context of constrained optimization, researchers have also proposed efficient sketching-based algorithms [38, 39, 40, 41, 42] under a similar notion of conic restricted strong-convexity.

## 1.3 This Work

In this paper we extend the theoretical framework for randomized first order methods established in [36] to design and analyse a structure-adaptive variant of Katyusha [23]. Our proposed method Rest-Katyusha is a restarted version of the original Katyusha method for non-strongly convex functions, where the restart period is determined by the modified restricted strong-convexity (RSC). The convergence analysis of Rest-Katyusha algorithm is provided, wherein we prove linear convergence up to a statistical accuracy with an accelerated convergence rate characterized by the RSC property.

Like all other accelerated gradient methods which require the explicit knowledge of strong-convexity parameter to achieve accelerated linear convergence, the vanilla Rest-Katyusha method also need to explicitly know the RSC parameter. We therefore propose a practical heuristic (adaptive Rest-Katyusha) which estimates the RSC parameter on the fly and adaptively tune the restart period, and we show that this adaptive scheme mimics the convergence behavior of the vanilla Rest-Katyusha. Finally we validate the effectiveness of our approach via numerical experiments.

## 2   Restarted Katyusha Algorithm

The Katyusha algorithm [23] listed in Algorithm 1 is an accelerated stochastic variance-reduced gradient method extended from the linear-coupling framework for constructing accelerated methods [43]. Its main loop (denoted as $\mathcal{A}$ in Algorithm 1) at iteration $s$ is described as the following:

For $\quad k = 0, 1, 2, ..., m$

$$
\begin{aligned}
x_{k+1} &= \theta z_k + \tfrac{1}{2}\hat{x}^s + (\tfrac{1}{2} - \theta)y_k; && \rightarrow \text{Linear coupling} \\
\nabla_{k+1} &= \nabla f(\hat{x}^s) + \nabla f_i(x_{k+1}) - \nabla f_i(\hat{x}^s); && \rightarrow \text{Variance reduced stochastic gradient} \\
z_{k+1} &= \arg\min_z \tfrac{3\theta L}{2}\|z - z_k\|_2^2 + \langle \nabla_{k+1}, z \rangle + \lambda g(z); && \rightarrow \text{Proximal mirror descent} \\
y_{k+1} &= \arg\min_y \tfrac{3L}{2}\|y - x_{k+1}\|_2^2 + \langle \nabla_{k+1}, y \rangle + \lambda g(y); && \rightarrow \text{Proximal gradient descent}
\end{aligned}
$$

The output sequence of $\mathcal{A}$ is defined as $\hat{x}^{s+1} = \frac{1}{m}\sum_{j=1}^{m} y_j$, $y^{s+1} = y_m$, $z^{s+1} = z_m$. It is one of the state-of-the-art methods for empirical risk minimization and matches the complexity lower-bound for minimizing smooth-convex finite-sum functions, proven by Lan and Zhou [21]. Most notably, it is a primal method which directly[1] accelerates stochastic variance-reduction methods. To achieve acceleration in the sense of Nesterov, Katyusha introduces the three-point coupling strategy which includes a combination of Nesterov's momentum and a stabilizing negative momentum which cancels the effect of noisy updates due to stochastic gradients. However, its accelerated linear convergence is only established when the regularization term $g(x)$ is strongly-convex, and fails to benefit from the strong convexity from the data-fidelity term [31], or the intrinsic restricted strong-convexity [36].

| **Algorithm 1** Katyusha $(x^0, m, S, L)$ | **Algorithm 2** Rest-Katyusha $(x^0, \mu_c, S_0, \beta, T, L)$ |
| --- | --- |
| **Initialize:** $y^0 = z^0 = \hat{x}^0$; <br> **for** $s = 0, \dots, S-1$ **do** <br> $\quad \theta \leftarrow \frac{2}{s+4}$, calculate $\nabla f(\hat{x}^s)$, <br><br> $\quad (\hat{x}^{s+1}, y^{s+1}, z^{s+1})$ <br> $\quad = \mathcal{A}(\hat{x}^s, y^s, z^s, \theta, \nabla f(\hat{x}^s), m)$ <br> **end for** <br> **Output:** $\hat{x}^S$ | **Initialize:** $m = 2n$, $S = \left\lceil \beta\sqrt{32 + \frac{24L}{m\mu_c}} \right\rceil$; <br> First stage —- warm start: <br> $x^1 =$ Katyusha $(x^0, m, S_0, L)$ <br> Second stage —- exploit the restricted strong-convexity via periodic restart: <br> **for** $t = 1, ..., T$ **do** <br> $\quad x^{t+1} =$ Katyusha $(x^t, m, S, L)$ <br> **end for** |

**Restart to rescue:** it is well-known that if the cost function $F(x)$ is $\mu$-strongly convex, one can periodically restart the accelerated full gradient method [16], and improve it from a sublinear convergence rate $F(x^k) - F^\star \leq \frac{4L\|x^0 - x^\star\|_2^2}{k^2}$ to a linearly convergent algorithm. For instance if we set $k = \left\lceil 4\sqrt{L/\mu} \right\rceil$, then one can show that the suboptimality can be reduced by $\frac{1}{4}$:

$$
F(x^k) - F^\star \leq \frac{4L\|x^0 - x^\star\|_2^2}{k^2} \leq \frac{4L[F(x^0) - F^\star]}{\mu k^2} \leq \frac{1}{4}[F(x^0) - F^\star]. \tag{4}
$$

Then we can recursively apply this statement (algorithmically speaking, we restart the algorithm every $\left\lceil 4\sqrt{L/\mu} \right\rceil$ iteration), and only $k \geq \left\lceil 4\sqrt{\frac{L}{\mu}} \right\rceil \log_4 \frac{1}{\delta}$ iterations are needed to make $F(x^k) - F^\star \leq \delta$, and hence an accelerated linear rate is achieved. The restart scheme has been recently applied to improve the convergence of the accelerated coordinate descent method [45, 30] and accelerated variance-reduced dual-averaging method [25] for strongly-convex functions.

Inspired by Nesterov [16] we first propose the Katyusha method with periodic restarts, and meanwhile demonstrate that when the restart period is appropriately chosen, the proposed method is able to exploit the restricted strong-convexity property to achieve an accelerated linear convergence, even when the cost function itself is not strongly-convex. We propose to warm start the algorithm prior to the periodic restart stage, by running the Katyusha algorithm for a number of epochs, which in theory should be proportional to the suboptimality of the starting point $x^0$. We present our Rest-Katyusha method as Algorithm 2.

## 3 Convergence Analysis of Rest-Katyusha

### 3.1 Generic Assumptions

We start by listing out the assumptions we may engage with in our analysis:

**A. 1.** *(Decomposable regularizer) [34] Given a orthogonal subspace pair $(\mathcal{M}, \mathcal{M}^\perp)$ in $\mathbb{R}^d$, $g(.)$ is decomposable which means:*

$$g(a + b) = g(a) + g(b), \forall a \in \mathcal{M}, b \in \mathcal{M}^\perp. \tag{5}$$

In this paper we focus on cases where the regularizer is decomposable, which includes many popular regularization which can enforce low-dimensional structure, such as $\ell_1$ norm, $\ell_{1,2}$ norm and nuclear norm penalty. The subspace $\mathcal{M}$ is named the *model subspace*, while its orthogonal complement $\mathcal{M}^\perp$ is called *perturbation subspace*. Similar notion of decomposition would extend the scope of this work to more general gauge functions $g(.)$, such as the so-called *analysis priors*, e.g., total variation regularization (for more details see Vaiter et al. [46]).

**A. 2.** *(Restricted strong convexity) [34] The function $f(.)$ satisfies restricted strong convexity with respect to $g(.)$ with parameters $(\gamma, \tau)$ if the following inequality holds true:*

$$f(x) - f(x^\star) - \langle \nabla f(x^\star), x - x^\star \rangle \geq \frac{\gamma}{2}\|x - x^\star\|_2^2 - \tau g^2(x - x^\star), \quad \forall x \in \mathbb{R}^d. \tag{6}$$

In [34], $\gamma$ is referred as the *lower curvature parameter*, while $\tau$ is named the *tolerance parameter*. It is clear that if $\tau = 0$, **A.2** reduces to usual strong-convexity assumption. While in the high-dimensional setting, the strong-convexity does not hold, but it has been shown in literature that such milder assumption of RSC does hold in many situations. This notion of RSC distinguishes from other forms of weak strong-convexity assumption based on the Polyak-Lojasiewicz inequality [47] for the purpose of this work, because it encodes the direction-restricting effect of the regularization, and hence has been shown to have a direct connection with the low-dimensional structure of $x^\star$. Next we define a crucial property for our structure-driven analysis, which is called the *subspace compatibility*:

**Definition 3.1.** *[34] With predefined $g(x)$, we define the subspace compatibility of a model subspace $\mathcal{M}$ as:*

$$\Phi(\mathcal{M}) := \sup_{v \in \mathcal{M} \setminus \{0\}} \frac{g(v)}{\|v\|_2}, \tag{7}$$

*when $\mathcal{M} \neq \{0\}$.*

The subspace compatibility $\Phi(\mathcal{M})$ captures the model complexity of subspace $\mathcal{M}$. For example if $g(.) = \|.\|_1$ and $\mathcal{M}$ is a subspace which is on a $s$-sparse support in $\mathbb{R}^d$, we will have $\Phi(\mathcal{M}) = \sqrt{s}$.

**A. 3.** *Each $f_i(.)$ has L-Lipschitz continuous gradient:*

$$\|\nabla f_i(x) - \nabla f_i(x')\|_2 \leq L\|x - x'\|_2, \forall x, x' \in \mathbb{R}^d. \tag{8}$$

This form of smoothness assumption is classic for variance-reduced stochastic gradient methods.

**A. 4.** *The regularization parameter $\lambda$ and $x^\dagger$ satisfies:*

$$\lambda \geq (1 + \frac{1}{c})g^*(\nabla f(x^\dagger)), \tag{9}$$

*with constant $c \geq 1$.*

Assumption **A.4** with the choice of $c = 1$ is the fundamental assumption of the analytical framework developed by Negahban et al. [3]. We relax the requirement to $c \geq 1$ for more general results. It is seemly a sophisticated and demanding assumption but indeed is reasonable and suits well the purpose of this work, which is to develop fast algorithms to speedily solve structured problems (which is always the result of sufficient regularization). Moreover, recall that the goal of finding the solution $x^\star$ via optimizing the regularized empirical risk is to get a meaningful approximation of the true parameter $x^\dagger$ which is the unique minimizer of the population risk. Especially in the high dimensional setting where $d > n$, the choice of regularization is rather important since there is no good control over the statistical error $\|x^\dagger - x^\star\|$ for an arbitrarily chosen $\lambda$. Because of this issue, in this work we only focus on the "meaningful" regularized ERM problems which are able to provide trustworthy approximation. Similar to **A.4**, Negahban et al.[3] has shown that $\lambda \geq 2g^*(\nabla f(x^\dagger))$ provides a sufficient condition to bound the statistical error $\|x^\star - x^\dagger\|_2^2$:

**Proposition 3.2.** *[3, Theorem 1, informal] Under **A.1**, **A.2**, **A.4**, with $c = 1$, if furthermore the curvature parameter $\gamma$, tolerance parameter $\tau$ and the subspace compatibility $\Phi(\mathcal{M})$ satisfy $\tau\Phi^2(\mathcal{M}) < \frac{\gamma}{64}$, then for any optima $x^\star$, the following inequality holds:*

$$\|x^\star - x^\dagger\|_2^2 \leq O\left(\frac{\lambda^2}{\gamma^2}\Phi^2(\mathcal{M}) + \frac{\lambda}{\gamma}g(x^\dagger_{\mathcal{M}^\perp})\right), \tag{10}$$

*where $O(.)$ hides deterministic constants for the simplicity of notation.*

Such a bound reveals desirable properties of the regularized ERM when the range of $\lambda$ satisfies assumption **A.4**. For instance, if $x^\dagger$ is the $s$-sparse ground truth vector of a noisy linear measurement system $y = Ax^\dagger + w$, where $w$ denotes the zero-mean sub-Gaussian noise (with variance $\sigma^2$) and the measurement matrix $A$ satisfies a certain restricted eigenvalue condition [3, 48], and we use a Lasso estimator $x^\star \in \arg\min_x \frac{1}{2n}\|Ax - y\|_2^2 + \lambda\|x\|_1$. In such case, let $\mathcal{M}$ be a subspace in $\mathbb{R}^d$ on $s$-sparse support where $x^\dagger \in \mathcal{M}$ and hence $g(x^\dagger_{\mathcal{M}^\perp}) = 0$, this proposition implies that:

$$\|x^\star - x^\dagger\|_2^2 \leq O\left(\frac{\lambda^2 s}{\gamma^2}\right) \approx O\left(\frac{\sigma^2}{\gamma^2}\frac{s\log d}{n}\right), \tag{11}$$

which implies the optimal convergence of the statistical error in terms of sample size and dimension for M-estimators. The details of this claim are presented in [3, Corollary 2].

## 3.2 Main Results

Base on the assumption of the restricted strong convexity on $f(.)$ w.r.t $g(.)$, and also with the definition of subspace compatibility, one can further derive a more expressive form of RSC, which is named *Effective RSC* [34] which has a directly link to the structure of solution.

**Lemma 3.3.** *(Effective RSC) [36, Lemma 3.3] Under **A.1**, **A.2**, **A.4**, while $x$ satisfies $F(x) - F(x^\star) \leq \eta$ for a given value $\eta > 0$ and any minimizer $x^\star$, with $\varepsilon := 2\Phi(\mathcal{M})\|x^\dagger - x^\star\|_2 + 4g(x^\dagger_{\mathcal{M}^\perp})$ we have:*

$$F(x) - F^\star \geq \mu_c\|x - x^\star\|_2^2 - 2\tau(1+c)^2 v^2, \tag{12}$$

*where $\mu_c = \frac{\gamma}{2} - 8\tau(1+c)^2\Phi^2(\mathcal{M})$ and $v = \frac{\eta}{\lambda} + \varepsilon$.*

Here we refer $\mu_c$ as the effective restricted strong convexity parameter, which will provide us a direct link between the convergence speed of an algorithm and the low-dimensional structure of the solution. Note that this lemma relaxes the condition on $\lambda$ in [34, Lemma 11], which is restricted to $c = 1$. Our main theorem is presented as the following:

**Theorem 3.4.** *Under **A.1 - 4**, if further **A.2** holds with parameter $(\gamma, \tau)$ such that $\tau\Phi^2(\mathcal{M}) < \frac{\gamma}{16(1+c)^2}$, denote $\varepsilon := 2\Phi(\mathcal{M})\|x^\dagger - x^\star\|_2 + 4g(x^\dagger_{\mathcal{M}^\perp})$, $\mathcal{D}(x^0, x^\star) := 16(F(x^0) - F^\star) + \frac{6L}{n}\|x^0 - x^\star\|_2^2$, $\mu_c = \frac{\gamma}{2} - 8\tau(1+c)^2\Phi^2(\mathcal{M})$, if we run Rest-Katyusha with $S_0 \geq \left\lceil \left(1 + \frac{2}{\rho\lambda}\right)\sqrt{\frac{6L\tau(1+c)^2\mathcal{D}(x^0, x^\star)}{8n\mu_c + 3L}} \right\rceil$, $S = \left\lceil \beta\sqrt{32 + \frac{12L}{n\mu_c}} \right\rceil$ with $\beta \geq 2$, then the following inequality holds:*

$$\mathbb{E}[F(x^{T+1}) - F^\star] \leq \max\left\{\varepsilon, \left(\frac{1}{\beta^2}\right)^T \frac{\mathcal{D}(x^0, x^\star)}{(S_0 + 3)^2}\right\} \tag{13}$$

*with probability at least $1 - \rho$.*

**Corollary 3.5.** *Under the same assumptions, parameter choices and notations as Theorem 3.4, the total number of stochastic gradient evaluations required by Rest-Katyusha to get an $\delta > \varepsilon$-accuracy is:*

$$O\left(n + \sqrt{\frac{nL}{\mu_c}}\right)\log\frac{1}{\delta} + O(n)S_0. \tag{14}$$

**Proof technique.** We extend the proof technique of Agarwal et al. [34] to the proximal gradient descent and also Qu and Xu [35] for SVRG which are both based on applying induction statements to roll up the residual term of (12) which is the second term at the RHS. The complete proofs of Theorem 3.4 and Corollary 3.5 can be found in the supplementary material.

**Accelerated linear convergence.** Under the RSC assumption, Theorem 3.4 and Corollary 3.5 demonstrate a local accelerated linear convergence rate of Rest-Katyusha up to a statistical accuracy $\delta > \varepsilon$. We derive this result based on extending the framework provided by Agarwal et al. [34], by which they established fast structure-dependent linear convergence of proximal gradient descent method up to a statistical accuracy of $\delta > \varepsilon$. To the best of our knowledge, this is the first structure-adaptive convergence result for an accelerated incremental gradient algorithm. Note that, this result can be trivially extended to a global accelerated linear convergence result (with $S_0 = S$) with the same setting of Agarwal et al. [34] where a side constraint $g(x) \leq R$ for some radii $R$ is added to restrict the early iterations with additional re-projections unto this constraint set[2]. Start from the objective-gap convergence result (13), with some additional algebra one can easily derive the accelerated linear convergence on the iterate (optimization variable) using the RSC condition as well.

**Structure-adaptive convergence.** The effective RSC $\mu_c = \frac{\gamma}{2} - 8\tau(1 + c)^2\Phi^2(\mathcal{M})$ links the convergence speed of Rest-Katyusha with the intrinsic low-dimensional structure of the solution which is due to the regularization. For instance, if $F(x) := \frac{1}{2n}\|Ax - b\|_2^2 + \lambda\|x\|_1$, $\|x^\star\|_0 = s$ and (A.4) holds $c = 1$, then we have $\mu_c = \frac{\gamma}{2} - 32\tau s$, meanwhile for a wide class of random design matrices we have $\tau = O(\frac{\log d}{n})$ and $\gamma > 0$. More specifically, if the rows of the random design matrix $A$ are drawn i.i.d. from $\mathcal{N}(0, \Sigma)$ with covariance matrix $\Sigma \in \mathbb{R}^{d \times d}$ which has largest singular value $r_{\max}(\Sigma)$ and smallest singular value $r_{\min}(\Sigma)$, then $\gamma \geq \frac{r_{\min}(\Sigma)}{16}$ and $\tau \leq r_{\max}(\Sigma)\frac{81\log d}{n}$ with high probability as shown by Raskutti et al. [48].

**High probability statement.** Since our proofs utilize the effective RSC which holds in a neighborhood of $x^\star$ as demonstrated in Lemma 3.3, we need to bound the functional suboptimality $F(x^t) - F^\star$ in the worst case instead of in expectation. Hence inevitably the Markov inequality has to be applied to provide the convergence statement with high probability (details can be found in the main proof).

**Optimizing the choice of $\beta$.** Theorem 3.4 shows that the complexity of the main loop of Rest-Katyusha is $\left\lceil \beta\sqrt{32 + 12L/(n\mu_c)} \right\rceil \log_{\beta^2}\frac{1}{\delta}$, which suggest a trade-off between the choice of $\beta$ and the total computation. With some trival computation one can derive that in theory the best choice of $\beta$ is exactly the Euler's number ($\approx 2.7$). Numerically, we observe that slightly larger choice of $\beta$ often provides better performance in practice (illustrative examples can be found in supplemental material).

## 4   Adaptive Rest-Katyusha

Motivated by the theory above, we further propose our practical adaptive restart heuristic of Rest-Katyusha which is able to estimate the effective RSC on the fly. Based on the convergence theory, we observe that, with the choice of restart period $S = \left\lceil \beta\sqrt{32 + 12L/(n\mu_0)} \right\rceil$ with a conservative estimate $\mu_0 \leq \mu_c$, then we are always guaranteed to have:

$$\mathbb{E}_{\xi_t \setminus \xi_{t-1}}[F(x^{t+1}) - F^\star] \leq \frac{1}{\beta^2}[F(x^t) - F^\star], \tag{15}$$

due to the fact that an underestimation of the RSC will leads to a longer restart period that we actually need[3]. The intuition behind our adaptive restart heuristic is: if we overestimate $\mu_c$, the above inequality will be violated. Hence an adaptive estimation of $\mu_c$ can be achieved via a convergence speed check. However the above inequality cannot be evaluated directly in practice since it is in expectation and demands the knowledge of $F^\star$. In [29, Prop. 4], it has been shown that with the composite gradient map:

$$\mathcal{T}(x) = \arg\min_q \frac{L}{2}\|x - q\|_2^2 + \langle \nabla f(x), q - x \rangle + \lambda g(q), \tag{16}$$

the value of $F(x) - F^\star$ can be lower bounded:

$$F(x) - F^\star \geq O\|\mathcal{T}(x) - x\|_2^2, \tag{17}$$

and also it can be approximately upper bounded by $O(\|\mathcal{T}(x) - x\|_2^2)$ if local quadratic growth is assumed, which reads:

$$\exists \alpha > 0, \ r > 0, \ F(x) - F^\star \geq \alpha\|x - x^\star\|_2^2, \forall x \ s.t. \ \|x - x^\star\|_2^2 < r. \tag{18}$$

Hence in our adaptive restart heuristic we check the convergence speed via evaluating the composite gradient map at the snapshot points where full gradients have already been calculated. Because of this, the only main additional computational overhead of this adaptive restart scheme is the proximal operation of $g(.)$ at the restart points.

---

**Algorithm 3** Adaptive Rest-Katyusha $(x^0, \mu_0, S_0, \beta, T, L)$

---

**Initialize:** Epoch length $m = 2n$; Initial restart period $S = \left\lceil \beta\sqrt{32 + \frac{12L}{n\mu_0}} \right\rceil$;

$x^1$ = Katyusha $(x^0, m, S_0, L)$
Calculate the composite gradient map:
$\mathcal{T}(x^1) = \arg\min_x \frac{L}{2}\|x - x^1\|_2^2 + \langle \nabla f(x^1), x - x^1 \rangle + \lambda g(x)$.
**for** $t = 1, \ldots, T$ **do**
    $x^{t+1}$ = Katyusha $(x^t, m, S, L)$
    ——Track the convergence speed via the composite gradient maps:
        $\mathcal{T}(x^{t+1}) = \arg\min_x \frac{L}{2}\|x - x^{t+1}\|_2^2 + \langle \nabla f(x^{t+1}), x - x^{t+1} \rangle + \lambda g(x)$.
    —— Update the estimate of RSC and adaptively tune the restart period
        **if** $\|\mathcal{T}(x^{t+1}) - x^{t+1}\|_2^2 \leq \frac{1}{\beta^2}\|\mathcal{T}(x^t) - x^t\|_2^2$
        **then** $\mu_0 \leftarrow 2\mu_0$, **else** $\mu_0 \leftarrow \mu_0/2$. $S = \left\lceil \beta\sqrt{32 + \frac{12L}{n\mu_0}} \right\rceil$
**end for**

---

The adaptive Rest-Katyusha method is presented in Algorithm 3. We highlight the heuristic estimating procedure for RSC parameter in the orange lines, which is additional to the original Katyusha algorithm. The algorithm start with an initial guess $\mu_0$ and correspondingly the restart period $S$, meanwhile we calculate the composite gradient map $\mathcal{T}(x^1)$ at $x^1$ and record the value of $\|\mathcal{T}(x^1) - x^1\|_2^2$ which we use as the estimation of $F(x^1) - F^\star$ (and so on). Then after $S$ outer-loops, we restart the algorithm and meanwhile and evaluate again the composite gradient map. If $\|\mathcal{T}(x^2) - x^2\|_2^2 \geq \frac{1}{\beta^2}\|\mathcal{T}(x^1) - x^1\|_2^2$, then we suspect that the RSC parameter has been overestimated, and hence we reduce $\mu_0$ by a half, otherwise we double the estimation. We also update the restart period by $S = \left\lceil \beta\sqrt{32 + 12L/(n\mu_0))} \right\rceil$ with the modified $\mu_0$. The forthcoming iterations follow the same updating rule as described.

## 5   Numerical Experiments

In this section we describe our numerical experiments on our proposed algorithm Rest-Katyusha (Alg.2) and also the adaptive Rest-Katyusha (Alg.3). We focus on the Lasso regression task:

$$x^\star \in \arg\min_{x \in \mathbb{R}^d} \left\{ F(x) := \frac{1}{2n}\|Ax - b\|_2^2 + \lambda\|x\|_1 \right\}. \tag{19}$$

To enforce sparsity on regression parameter we use the $\ell_1$ penalty with various degrees of regularization parameters chosen from the set $\lambda \in \{1 \times 10^p, 2 \times 10^p, 5 \times 10^p, p \in \mathbb{Z}\}$. For comparison, the performance of (proximal) SVRG and the original Katyusha method is also shown in the plots. We run all the algorithms with their theoretical step sizes for Madelon and REGED dataset, while for RCV1 dataset we adopt minibatch scheme for all the algorithms and grid-search the step sizes which optimize these algorithms' performance.

Table 1: Datasets for the Experiments and Minibatch Size Choice for the Algorithms

| DATA SET | SIZE $(n, d)$ | MINIBATCH | REF. |
|---|---|---|---|
| **(A)** MADELON | (2000, 500) | 1 | [49] |
| **(B)** RCV1 | (20242, 47236) | 80 | [49] |
| **(C)** REGED | (500, 999) | 1 | [50] |

Figure 1: Lasso Experiments on **(A)** Madelon and **(B)** RCV1

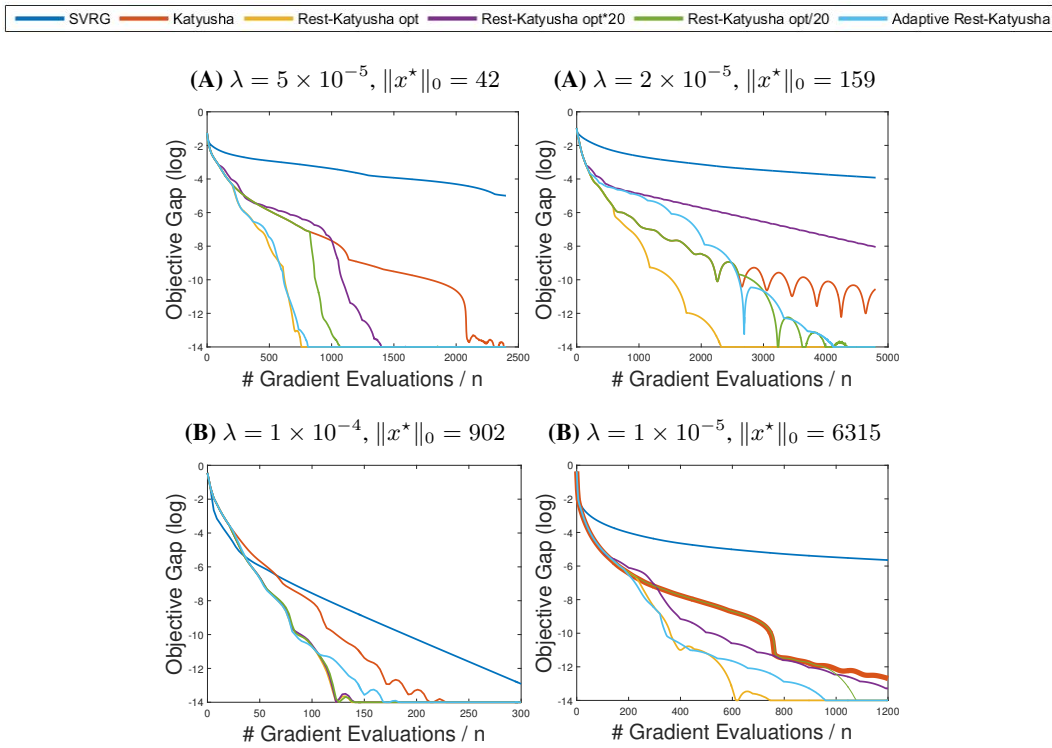

In all our experiments we set $\beta = 5$ and $S_0 = S$ for convenience. We first do a grid-search on the estimate of $\mu_c$ for Rest-Katyusha which provides the best convergence performance, and denote it as "Rest-Katyusha opt" in the plots. Meanwhile we also run Rest-Katyusha with RSC estimation which is 20 times larger or smaller than the optimal one, where we denote as "Rest-Katyusha opt*20" and "Rest-Katyusha opt/20" respectively. At the 5th plot in Figure 1 the curves for Rest-Kat opt, opt*20 and opt/20 are indistinguishable which shows that in these particular experiments their performance are almost identical. For the adaptive Rest-Katyusha we fix our starting estimate of $\mu_c$ as $10^{-5}$ throughout all the experiments.

From these experiments we observe that as our theory has predicted, the Rest-Katyusha achieves accelerated linear convergence even when there is no explicit strong-convexity in the cost function (RCV1 and REGED dataset), and the convergence speed has a direct relationship with the sparsity of solution. For the lasso experiments while the solution is sparser, the linear convergence speed of Rest-Katyusha indeed become faster. Meanwhile when we run Rest-Katyusha with an inaccurate RSC estimate, we still observe a compromised linear convergence, as predicted by our theory. In all the experiments, we have observe that the adaptive Rest-Katyusha indeed achieves a good estimation

Figure 2: Lasso Experiments on **(C)** REGED Dataset

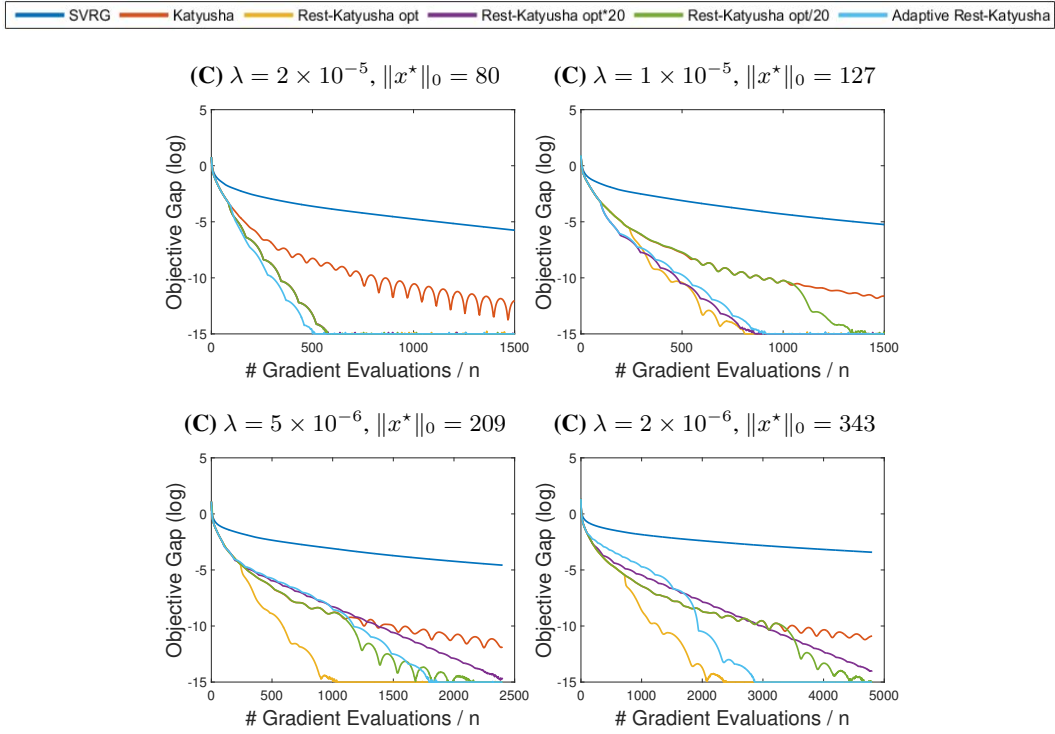

of the RSC parameter and properly adapts the choice of restart period automatically on the fly, hence its performance is often comparable with the best tuned Rest-Katyusha. As the experimental results shown in [34, 35, 36], the linear convergence we observe is towards an arbitrary accuracy instead of a threshold nearby the solution. This conservative aspect of the theory is inherently due to the artifact of the RSC framework [34] and we include the extension for arbitrary accuracy regime as our future work. *We also include additional experimental results in the supplemental materials.*

# 6 Conclusion

We developed a restart variant of the Katyusha algorithm for regularized empirical risk minimization tasks, which is provably able to actively exploit the intrinsic low-dimensional structure of the solution for the acceleration of convergence. Based on the convergence result we further constructed an adaptive restart heuristic which aimed at estimating the RSC parameter on the fly and adaptively tune the restart period. The efficiency of this approach is validated through numerical experiments. In future work, we aim to develop more refined and provably-good adaptive restart schemes for Rest-Katyusha algorithm to further exploit the solution's structure for acceleration.

# 7 Acknowledgements

JT, FB, MG and MD would like to acknowledge the support from H2020-MSCA-ITN Machine Sensing Training Network (MacSeNet), project 642685; ERC grant SEQUOIA, project 724063; EPSRC Compressed Quantitative MRI grant, number EP/M019802/1; and ERC Advanced grant, project 694888, C-SENSE, respectively. MD is also supported by a Royal Society Wolfson Research Merit Award. JT would like to thank Damien Scieur and Vincent Roulet for helpful discussions during his research visit in SIERRA team.

## Footnotes

[1]On the other hand, one can indirectly accelerate SVRG/SAGA via Catalyst [44].

[2]In [34], a side constraint is manually added to the regularized ERM problem, hence in their setting, the effective restricted strong-convexity is valid globally. They provide global linear convergence result of proximal gradient descent (with additional re-projection steps) at a cost of additional side-constraints.

[3]An inaccurate estimate of the RSC will lead to a compromised convergence rate. Detailed discussion and analysis of Rest-Katyusha with a rough RSC estimate can be found in the Appendix.

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
