[Supplementary Material]

# Rest-Katyusha: Exploiting the Solution's Structure via Scheduled Restart Schemes
## *(Supplementary Material)*

**Junqi Tang**
School of Engineering
University of Edinburgh, UK
J.Tang@ed.ac.uk

**Mohammad Golbabaee**
Department of Computer Science
University of Bath, UK
M.Golbabaee@bath.ac.uk

**Francis Bach**
INRIA - ENS
PSL Research University, France
Francis.Bach@inria.fr

**Mike Davies**
School of Engineering
University of Edinburgh, UK
Mike.Davies@ed.ac.uk

In this supplementary material we include in section A the proof of our main result which establishes the structure-adaptive and accelerated linear convergence rate for Rest-Katyusha algorithm. Further, in section B we also extend our analysis to the case where we underestimate the RSC parameter. We also provide an additional numerical result for testing the convergence rate of Rest-Katyusha with different choices of the input parameter $\beta$.

## A    Proof of Theorem 3.4 and Corollary 3.5

We first state the convergence result for Katyusha algorithm for non-strongly convex functions:

**Lemma A.1.** *[1, Theorem 4.1] Under A.3, starting at $x^0$, with epoch length $m = 2n$, denote $\mathcal{D}(x^0, x^\star) := 16(F(x^0) - F^\star) + \frac{6L}{n}\|x^0 - x^\star\|_2^2$, the s-th snapshot point $\hat{x}^s$ of Katyusha algorithm satisfies:*

$$\mathbb{E}[F(\hat{x}^s)] - F^\star \leq \frac{\mathcal{D}(x^0, x^\star)}{(s+3)^2}. \tag{1}$$

Now based on the inequality of effective RSC by Lemma 3.3 in the main text we are able to provide the proof of our main result.

*Proof.* At each iteration, the algorithm chooses an index $i$ uniformly at random to perform the calculation of one stochastic variance-reduced gradient. The update sequences $y_{k+1}$ and $z_{k+1}$ within $t$-th outer-loop of Rest-Katyusha depend on the realization of the following random variable which we denote as $\xi_k^t$:

$$\xi_t = \{i_m^t, i_{m-1}^t, ..., i_1^t, i_0^t, i_m^{t-1}, ..., i_0^{t-1}, ..., i_m^0, ..., i_0^0\}, \tag{2}$$

and for the randomness within a single outer-loop of Rest-Katyusha we specifically denote $\xi_t \backslash \xi_{t-1}$ as

$$\xi_t \backslash \xi_{t-1} = \{i_m^t, i_{m-1}^t, ..., i_1^t, i_0^t\}. \tag{3}$$

According to Lemma A.1, setting $m = 2n$, for the first stage $t = 0$:

$$\mathbb{E}_{\xi_0}\left[F(x^1)\right] - F^\star \leq \epsilon_1 := \frac{4}{n(S_0 + 3)^2}\left[4n\left(F(x^0) - F^\star\right) + \frac{3L}{2}\|x^0 - x^\star\|_2^2\right].$$

Then, applying Markov's inequality, with probability at least $1 - \frac{\rho}{2}$ we have:

$$F(x^1) - F^\star \leq \frac{2}{\rho}\epsilon_1. \tag{4}$$

Then we define three sequences $\epsilon_t$, $\rho_t$ and $v_t$: $\epsilon_{t+1} = \frac{1}{\beta^2}\epsilon_t > \varepsilon$, $\rho_{t+1} = \frac{1}{\beta}\rho_t$ (with $\rho_1 := \rho$), $v_t = \frac{2\epsilon_t}{\lambda\rho_t} + \varepsilon$. Next we use and induction argument to upper bound $\mathbb{E}_{\xi_{t-1}}F(x^t) - F^\star$.

**Induction step 1:** We turn to the first iteration of the second stage, note that due to the effective RSC, we can write:

$$\|x - x^\star\|_2^2 \leq \frac{1}{\mu_c}\left[F(x) - F^\star + 2\tau(1+c)^2v^2\right], \tag{5}$$

hence we can have the following:

$$
\begin{aligned}
\mathbb{E}_{\xi_1 \setminus \xi_0}[F(x^2) - F^\star] &\leq \frac{16}{(S+3)^2}[F(x^1) - F^\star] + \frac{6L}{n\mu_c(S+3)^2}\left[F(x^1) - F^\star + 2\tau(1+c)^2v_1^2\right] \\
&\leq \frac{16 + \frac{6L}{n\mu_c}}{(S+3)^2}[F(x^1) - F^\star] + \frac{12L\tau(1+c)^2}{n\mu_c(S+3)^2}v_1^2,
\end{aligned}
$$

and then we take expectation over $\xi_0$ we have:

$$
\begin{aligned}
\mathbb{E}_{\xi_1}[F(x^2) - F^\star] &\leq \frac{16 + \frac{6L}{n\mu_c}}{(S+3)^2}\mathbb{E}_{\xi_0}[F(x^1) - F^\star] + \frac{12L\tau(1+c)^2}{n\mu_c(S+3)^2}v_1^2 \\
&= \frac{16 + \frac{6L}{n\mu_c}}{(S+3)^2}\epsilon_1 + \frac{12L\tau(1+c)^2}{n\mu_c(S+3)^2}\left(\frac{2\epsilon_1}{\rho\lambda} + \varepsilon\right)^2 \\
&\leq \frac{16 + \frac{6L}{n\mu_c}}{(S+3)^2}\epsilon_1 + \frac{12L\tau(1+c)^2}{n\mu_c(S+3)^2}\left(\frac{2\epsilon_1}{\rho\lambda} + \epsilon_1\right)^2.
\end{aligned}
$$

then we set:

$$\frac{12L\tau(1+c)^2}{n\mu_c}\left[\left(\frac{2}{\rho\lambda} + 1\right)\epsilon_1\right]^2 \leq \left(16 + \frac{6L}{n\mu_c}\right)\epsilon_1, \tag{6}$$

equivalently:

$$\left(\frac{2}{\rho\lambda} + 1\right)^2\epsilon_1 \leq \frac{8n\mu_c + 3L}{6L\tau(1+c)^2}, \tag{7}$$

and denote $\mathcal{D}(x^0, x^\star) := 16[F(x^0) - F^\star] + \frac{6L}{n}\|x^0 - x^\star\|_2^2$, we have:

$$\epsilon_1 := \frac{\mathcal{D}(x^0, x^\star)}{(S_0 + 3)^2} \leq \frac{8n\mu_c + 3L}{6L\tau(1+c)^2(\frac{2}{\rho\lambda} + 1)^2}. \tag{8}$$

Hence in order to satisfy inequality (6), it is enough to set:

$$S_0 \geq \left\lceil\left(1 + \frac{2}{\rho\lambda}\right)\sqrt{\frac{6L\tau(1+c)^2\mathcal{D}(x^0, x^\star)}{8n\mu_c + 3L}}\right\rceil. \tag{9}$$

By this choice of $S_0$, according to inequality (6) we can write:

$$\mathbb{E}_{\xi_1}[F(x^2) - F^\star] \leq \frac{32 + \frac{12L}{n\mu_c}}{(S+3)^2}\epsilon_1, \tag{10}$$

to get $\mathbb{E}_{\xi_1}[F(x^2) - F^\star] \leq \frac{1}{\beta^2}\epsilon_1 = \epsilon_2$, it is enough to set:

$$S = \left\lceil\beta\sqrt{32 + \frac{12L}{n\mu_c}}\right\rceil \tag{11}$$

**Induction step 2:** For the $(t+1)$-th iteration, according to the induction hypothesis, we have $\mathbb{E}_{\xi_{t-1}} F(x^t) - F^\star \leq \frac{\epsilon_{t-1}}{\beta^2} = \epsilon_t$, and hence with probability $1 - \frac{\rho_t}{2}$ we have:

$$
\begin{aligned}
\mathbb{E}_{\xi_t}[F(x^{t+1}) - F^\star] &\leq \frac{16 + \frac{6L}{n\mu_c}}{(S+3)^2} \mathbb{E}_{\xi_{t-1}}[F(x^t) - F^\star] + \frac{12L\tau(1+c)^2}{n\mu_c(S+3)^2} v_t^2 \\
&= \frac{16 + \frac{6L}{n\mu_c}}{(S+3)^2} \epsilon_t + \frac{12L\tau(1+c)^2}{n\mu_c(S+3)^2} \left( \frac{2\epsilon_t}{\rho_t \lambda} + \varepsilon \right)^2 \\
&\leq \frac{16 + \frac{6L}{n\mu_c}}{(S+3)^2} \epsilon_t + \frac{12L\tau(1+c)^2}{n\mu_c(S+3)^2} \left( \frac{2\epsilon_t}{\rho_t \lambda} + \epsilon_t \right)^2.
\end{aligned}
$$

then we set:

$$
\frac{12L\tau(1+c)^2}{n\mu_c} \left[ \left( \frac{2}{\rho_t \lambda} + 1 \right) \epsilon_t \right]^2 \leq \left( 16 + \frac{6L}{n\mu_c} \right) \epsilon_t, \tag{12}
$$

equivalently:

$$
\left( \frac{2}{\rho_t \lambda} + 1 \right)^2 \epsilon_t \leq \frac{8n\mu_c + 3L}{6L\tau(1+c)^2}, \tag{13}
$$

Now because $\rho_t = \frac{1}{\beta} \rho_{t-1}$, $\epsilon_t = \frac{1}{\beta^2} \epsilon_{t-1}$, we have:

$$
\left( \frac{2}{\rho_t \lambda} + 1 \right)^2 \epsilon_t = \left( \frac{2}{\rho_{t-1}\lambda} + \frac{1}{\beta} \right)^2 \epsilon_{t-1} \leq \left( \frac{2}{\rho_{t-1}\lambda} + 1 \right)^2 \epsilon_{t-1} \leq \ldots \leq \left( \frac{2}{\rho\lambda} + 1 \right)^2 \epsilon_1. \tag{14}
$$

Hence by the same choice of $S_0$ given by (9), inequality (12) holds and consequently we can have:

$$
\mathbb{E}_{\xi_t}[F(x^{t+1}) - F^\star] \leq \frac{32 + \frac{12L}{n\mu_c}}{(S+3)^2} \epsilon_t, \tag{15}
$$

to get $\mathbb{E}_{\xi_t}[F(x^{t+1}) - F^\star] \leq \frac{1}{\beta^2} \epsilon_t = \epsilon_{t+1}$, it is enough to set:

$$
S = \left\lceil \beta \sqrt{32 + \frac{12L}{n\mu_c}} \right\rceil. \tag{16}
$$

Hence we finish the induction – by the choice of:

$$
S_0 \geq \left\lceil \left( 1 + \frac{2}{\rho\lambda} \right) \sqrt{\frac{6L\tau(1+c)^2 \mathcal{D}(x^0, x^\star)}{8n\mu_c + 3L}} \right\rceil, \quad S = \left\lceil \beta \sqrt{32 + \frac{12L}{n\mu_c}} \right\rceil, \tag{17}
$$

then we will have:

$$
\mathbb{E}_{\xi_t}[F(x^{t+1}) - F^\star] \leq \frac{\epsilon_t}{\beta^2} \tag{18}
$$

where $\epsilon_{t+1} = \frac{1}{\beta^2} \epsilon_t$ and $\epsilon_1 = \frac{\mathcal{D}(x^0, x^\star)}{(S_0+3)^2} = \frac{4}{n(S_0+3)^2} \left[ 4n \left( F(x^0) - F^\star \right) + \frac{3L}{2} \|x^0 - x^\star\|_2^2 \right]$, with probability $1 - \sum_{i=1}^{t} \frac{\rho_i}{2} \geq 1 - \frac{\rho}{2} \frac{\beta}{\beta-1} \geq 1 - \rho$ (since $\beta \geq 2$). Now we have finished the proof of Theorem 3.4.

**Proof of Corollary 3.5.** Finally we make a summary of this result for the proof of Corollary 3.5. First we write the number of snapshot point calculation we need to achieve $\mathbb{E}_{\xi_{t-1}} F(x^t) - F^\star \leq \delta$ at the second stage:

$$
N_s = \left\lceil \beta \sqrt{32 + \frac{12L}{n\mu_c}} \right\rceil \log_{\beta^2} \frac{F(x^1) - F^\star}{\delta}. \tag{19}
$$

When $\frac{2n\mu_c}{L} \leq \frac{3}{4}$, $N_s = O\left( \sqrt{\frac{L}{2n\mu_c}} \log \frac{F(x^1)-F^\star}{\delta} \right)$; when $\frac{2n\mu_c}{L} \geq \frac{3}{4}$, $N_s = O\left( \log \frac{F(x^1)-F^\star}{\delta} \right)$.

Hence it is enough to run $O\left( (1 + \sqrt{\frac{L}{2n\mu_c}}) \log \frac{F(x^1)-F^\star}{\delta} \right) \geq O\left( \max(1, \sqrt{\frac{L}{2n\mu_c}}) \log \frac{F(x^1)-F^\star}{\delta} \right)$

epochs. Since we set the epoch length $m = 2n$ and hence the number of stochastic gradient $\triangledown f_i(.)$ calculation is of $O(n)$. Therefore with some more straightforward calculation we conclude that the complexity of the Rest-Katyusha algorithm is:

$$N \geq O\left(n + \sqrt{\frac{nL}{\mu_c}}\right) \log \frac{\frac{1}{\rho(S_0+3)^2}\left[16(F(x^0) - F^\star) + \frac{6L}{n}\|x^0 - x^\star\|_2^2\right]}{\delta} + O(n)S_0. \quad (20)$$

$\square$

## B    Rest-Katyusha with an underestimation of $\mu_c$

It is generally not guaranteed that any accelerated stochastic variance-reduced gradient method designed for strongly-convex functions can be directly applied in our modified restricted strong-convexity setting, even when the RSC parameter can be exactly known. It is true that for strongly-convex functions with known strong-convexity parameter, that the convergence rates for a restarted version of non-strongly-convex accelerated gradient descent and the strongly-convex accelerated gradient descent are the same, and we believe that this may be the case for Katyusha as well if the objective is strongly-convex. However, it is still an open question for an objective which only satisfies restricted strong-convexity. One may heuristically replace our algorithm's second stage with the strongly-convex version of Katyusha and this seems to have a comparable result empirically for some datasets if the RSC is accurately given (this is necessary for this method). However, the Rest-Katyusha is superior to this alternative – (1) in terms of theory, as it is a provably convergent algorithm, (2) in terms of practice, Rest-Katyusha appears to be much more robust to the inaccurate estimation of RSC. This section we provide an analysis for Rest-Katyusha where we underestimate the RSC parameter.

We have already established the convergence result for Rest-Katyusha algorithm when it is restarted at a frequency $S = \left\lceil \beta\sqrt{32 + \frac{12L}{n\mu_c}} \right\rceil$, but in practice the effective RSC parameter $\mu_c$ is usually unknown and difficult to estimate accurately. We need to find some practical approaches to estimate $\mu_c$ and determine whether to restart or not on the fly. To lay down the basics, we now warm up with the analysis for Rest-Katyusha when only an underestimation of $\mu_c$ is given, to see how the convergence rate of the algorithm will change.

---
**Algorithm 1** Rest-Katyusha with a rough RSC estimate $(x^0, \mu_0, \beta, S_0, T, L)$
---
**Initialize:** $m = 2n$, $S = \left\lceil \beta\sqrt{32 + \frac{12L}{n\mu_0}} \right\rceil$;

$x^1$ = Katyusha $(x^0, m, S_0, L)$
**for** $t = 1, ..., T$ **do**
$\quad x^{t+1}$ = Katyusha $(x^t, m, S, L)$
**end for**
---

We present the rough RSC estimate version of Rest-Katyusha. The only difference is that the restart period has changed from $\left\lceil \beta\sqrt{32 + \frac{12L}{n\mu_c}} \right\rceil$ to $\left\lceil \beta\sqrt{32 + \frac{12L}{n\mu_0}} \right\rceil$, where $\mu_0$ is an rough (under-)estimate of the effective RSC constant $\mu_c$ and $\beta \geq 2$ is a constant which controls the robustness of possible overestimation. With this restart period, we are able to establish accelerated linear convergence result in the regime where $0 < \mu_0 < \frac{\beta^2}{4}\mu_c$. In other words, with this restart period, as long as $\mu_c$ is no more than $\beta^2/4$ times overestimated by $\mu_0$, the Rest-Katyusha is guaranteed to achieve accelerated linear convergence w.r.t. $\mu_0$.

**Theorem B.1.** *Under A.1 - 4, denote* $\varepsilon := 2\Phi(\mathcal{M})\|x^\dagger - x^\star\|_2 + 4g(x^\dagger_{\mathcal{M}^\perp})$, $\mathcal{D}(x^0, x^\star) := 16(F(x^0) - F^\star) + \frac{6L}{n}\|x^0 - x^\star\|_2^2$, $\mu_c = \frac{\gamma}{2} - 8\tau(1+c)^2\Phi^2(\mathcal{M})$, *and* $0 < \mu_0 < \frac{\beta^2}{4}\mu_c$, *with* $\beta \geq 2$, *if we run Rest-Katyusha with* $S_0 \geq \left\lceil \left(1 + \frac{2}{\rho\lambda}\right)\sqrt{2\tau(1+c)^2\mathcal{D}(x^0, x^\star)} \right\rceil$, $S = \left\lceil \beta\sqrt{32 + \frac{12L}{n\mu_0}} \right\rceil$, *then the following inequality holds:*

$$\mathbb{E}[F(x^{T+1}) - F^\star] \leq \max\left\{\varepsilon, \left(\frac{\mu_0}{\mu_c\beta^2}\right)^T \frac{\mathcal{D}(x^0, x^\star)}{(S_0 + 3)^2}\right\}, \quad (21)$$

*with probability at least $1 - \rho$.*

**Corollary B.2.** *Under the same assumptions, parameter choices and notations as Theorem B.1, the total number of stochastic gradient evaluation required by Rest-Katyusha to get an $\delta$-accuracy is:*

$$O\left(n + \sqrt{\frac{nL}{\mu_0}}\right) \log_{\frac{\beta^2 \mu_c}{\mu_0}} \frac{1}{\delta} + O(n)S_0, \tag{22}$$

*Proof.* At each iteration, the algorithm chooses an index $i$ uniformly at random to perform the calculation of one stochastic variance-reduced gradient. The update sequences $y_{k+1}$ and $z_{k+1}$ within $t$-th outer-loop of Rest-Katyusha depend on the realization of the following random variable which we denote as $\xi_k^t$:

$$\xi_t = \{i_m^t, i_{m-1}^t, ..., i_1^t, i_0^t, i_m^{t-1}, ..., i_0^{t-1}, ..., i_m^0, ..., i_0^0\}, \tag{23}$$

and for the randomness within a single outer-loop of Rest-Katyusha we specifically denote $\xi_t \backslash \xi_{t-1}$ as

$$\xi_t \backslash \xi_{t-1} = \{i_m^t, i_{m-1}^t, ..., i_1^t, i_0^t\}. \tag{24}$$

According to Lemma A.1, setting $m = 2n$, for first stage $t = 0$:

$$\mathbb{E}_{\xi_0}[F(x^1)] - F^\star \le \epsilon_1 := \frac{4}{n(S_0 + 3)^2}\left[4n\left(F(x^0) - F^\star\right) + \frac{3L}{2}\|x^0 - x^\star\|_2^2\right].$$

Then with probability at least $1 - \frac{\rho}{2}$ we have:

$$F(x^1) - F^\star \le \frac{2}{\rho}\epsilon_1. \tag{25}$$

Then we denote $\alpha = \frac{\mu_0}{\mu_c}$ and also define three sequences $\epsilon_t$, $\rho_t$ and $v_t$: $\epsilon_{t+1} = \frac{\alpha}{\beta^2}\epsilon_t$, $\rho_{t+1} = \frac{\sqrt{\alpha}}{\beta}\rho_t$ (with $\rho_1 := \rho$), $v_t = \frac{2\epsilon_t}{\lambda \rho_t} + \varepsilon$. Next we use and induction argument to upper bound $\mathbb{E}_{\xi_{t-1}} F(x^t) - F^\star$.

**Induction step 1:** We turn to the first iteration of the second stage, note that due to the effective RSC, we can write:

$$\|x - x^\star\|_2^2 \le \frac{1}{\mu_c}\left[F(x) - F^\star + 2\tau(1+c)^2 v^2\right], \tag{26}$$

hence we can have the following:

$$\begin{aligned}
\mathbb{E}_{\xi_1 \backslash \xi_0}[F(x^2) - F^\star] &\le \frac{16}{(S+3)^2}[F(x^1) - F^\star] + \frac{6L}{n\mu_c(S+3)^2}\left[F(x^1) - F^\star + 2\tau(1+c)^2 v_1^2\right] \\
&\le \frac{16 + \frac{6L}{n\mu_c}}{(S+3)^2}[F(x^1) - F^\star] + \frac{12L\tau(1+c)^2}{n\mu_c(S+3)^2}v_1^2,
\end{aligned}$$

and then we take expectation over $\xi_0$ we have:

$$\begin{aligned}
\mathbb{E}_{\xi_1}[F(x^2) - F^\star] &\le \frac{16 + \frac{6L}{n\mu_c}}{(S+3)^2}\mathbb{E}_{\xi_0}[F(x^1) - F^\star] + \frac{12L\tau(1+c)^2}{n\mu_c(S+3)^2}v_1^2 \\
&= \frac{16 + \frac{6L}{n\mu_c}}{(S+3)^2}\epsilon_1 + \frac{12L\tau(1+c)^2}{n\mu_c(S+3)^2}\left(\frac{2\epsilon_1}{\rho\lambda} + \varepsilon\right)^2 \\
&\le \frac{16 + \frac{6L}{n\mu_c}}{(S+3)^2}\epsilon_1 + \frac{12L\tau(1+c)^2}{n\mu_c(S+3)^2}\left(\frac{2\epsilon_1}{\rho\lambda} + \epsilon_1\right)^2.
\end{aligned}$$

then we set:

$$\frac{12L\tau(1+c)^2}{n\mu_c}\left[\left(\frac{2}{\rho\lambda} + 1\right)\epsilon_1\right]^2 \le \left(16 + \frac{6L}{n\mu_c}\right)\epsilon_1, \tag{27}$$

equivalently:

$$\left(\frac{2}{\rho\lambda} + 1\right)^2 \epsilon_1 \le \frac{8n\mu_c + 3L}{6L\tau(1+c)^2}, \tag{28}$$

and denote $\mathcal{D}(x^0, x^\star) := 16[F(x^0) - F^\star] + \frac{6L}{n}\|x^0 - x^\star\|_2^2$, we have:

$$\epsilon_1 := \frac{\mathcal{D}(x^0, x^\star)}{(S_0 + 3)^2} \leq \frac{8n\mu_c + 3L}{6L\tau(1 + c)^2(\frac{2}{\rho\lambda} + 1)^2}. \tag{29}$$

Hence in order to satisfy inequality (27), it is enough to set:

$$S_0 \geq \left\lceil \left(1 + \frac{2}{\rho\lambda}\right)\sqrt{2\tau(1 + c)^2 \mathcal{D}(x^0, x^\star)} \right\rceil \geq \left\lceil \left(1 + \frac{2}{\rho\lambda}\right)\sqrt{\frac{6L\tau(1 + c)^2 \mathcal{D}(x^0, x^\star)}{8n\mu_c + 3L}} \right\rceil. \tag{30}$$

By this choice of $S_0$, according to inequality (27) we can write:

$$\mathbb{E}_{\xi_1}[F(x^2) - F^\star] \leq \frac{32 + \frac{12L}{n\mu_c}}{(S + 3)^2}\epsilon_1, \tag{31}$$

to get $\mathbb{E}_{\xi_1}[F(x^2) - F^\star] \leq \frac{\alpha}{\beta^2}\epsilon_1 = \epsilon_2$, it is enough to set:

$$S = \left\lceil \beta\sqrt{32 + \frac{12L}{n\mu_0}} \right\rceil. \tag{32}$$

**Induction step 2:** For the $(t + 1)$-th iteration, according to the induction hypothesis, we have $\mathbb{E}_{\xi_{t-1}}F(x^t) - F^\star \leq \frac{\alpha\epsilon_{t-1}}{\beta^2} = \epsilon_t$, and hence with probability $1 - \frac{\rho_t}{2}$ we have:

$$\begin{aligned}
\mathbb{E}_{\xi_t}[F(x^{t+1}) - F^\star] &\leq \frac{16 + \frac{6L}{n\mu_c}}{(S + 3)^2}\mathbb{E}_{\xi_{t-1}}[F(x^t) - F^\star] + \frac{12L\tau(1 + c)^2}{n\mu_c(S + 3)^2}v_t^2 \\
&= \frac{16 + \frac{6L}{n\mu_c}}{(S + 3)^2}\epsilon_t + \frac{12L\tau(1 + c)^2}{n\mu_c(S + 3)^2}\left(\frac{2\epsilon_t}{\rho_t\lambda} + \varepsilon\right)^2 \\
&\leq \frac{16 + \frac{6L}{n\mu_c}}{(S + 3)^2}\epsilon_t + \frac{12L\tau(1 + c)^2}{n\mu_c(S + 3)^2}\left(\frac{2\epsilon_t}{\rho_t\lambda} + \epsilon_t\right)^2.
\end{aligned}$$

then we set:

$$\frac{12L\tau(1 + c)^2}{n\mu_c}\left[\left(\frac{2}{\rho_t\lambda} + 1\right)\epsilon_t\right]^2 \leq \left(16 + \frac{6L}{n\mu_c}\right)\epsilon_t, \tag{33}$$

equivalently:

$$\left(\frac{2}{\rho_t\lambda} + 1\right)^2\epsilon_t \leq \frac{8n\mu_c + 3L}{6L\tau(1 + c)^2}, \tag{34}$$

Now because $\rho_t = \frac{\sqrt{\alpha}}{\beta}\rho_{t-1}$, $\epsilon_t = \frac{\alpha}{\beta^2}\epsilon_{t-1}$, we have:

$$\left(\frac{2}{\rho_t\lambda} + 1\right)^2\epsilon_t = \left(\frac{2}{\rho_{t-1}\lambda} + \frac{\sqrt{\alpha}}{\beta}\right)^2\epsilon_{t-1} \leq \left(\frac{2}{\rho_{t-1}\lambda} + 1\right)^2\epsilon_{t-1} \leq \ldots \leq \left(\frac{2}{\rho\lambda} + 1\right)^2\epsilon_1. \tag{35}$$

Hence by the same choice of $S_0$ given by (30), inequality (33) holds and consequently we can have:

$$\mathbb{E}_{\xi_t}[F(x^{t+1}) - F^\star] \leq \frac{32 + \frac{12L}{n\mu_c}}{(S + 3)^2}\epsilon_t, \tag{36}$$

to get $\mathbb{E}_{\xi_t}[F(x^{t+1}) - F^\star] \leq \frac{\alpha}{\beta^2}\epsilon_t = \epsilon_{t+1}$, it is enough to set:

$$S = \left\lceil \beta\sqrt{32 + \frac{12L}{n\mu_c}} \right\rceil. \tag{37}$$

Hence we finish the induction – by the choice of:

$$S_0 \geq \left\lceil \left(1 + \frac{2}{\rho\lambda}\right)\sqrt{2\tau(1 + c)^2 \mathcal{D}(x^0, x^\star)} \right\rceil, \quad S = \left\lceil \beta\sqrt{32 + \frac{12L}{n\mu_0}} \right\rceil, \tag{38}$$

then we will have:

$$\mathbb{E}_{\xi_t}[F(x^{t+1}) - F^\star] \leq \frac{\alpha \epsilon_t}{\beta^2} \tag{39}$$

where $\epsilon_{t+1} = \frac{\alpha}{\beta^2}\epsilon_t$ and $\epsilon_1 = \frac{\mathcal{D}(x^0, x^\star)}{(S_0+3)^2} = \frac{4}{n(S_0+3)^2}\left[4n\left(F(x^0) - F^\star\right) + \frac{3L}{2}\|x^0 - x^\star\|_2^2\right]$, with probability $1 - \frac{1}{2}\sum_{i=1}^t \rho_i \geq 1 - \frac{\rho}{2}\frac{\beta}{\beta - \sqrt{\alpha}} \geq 1 - \rho$. Now we have finished the proof of Theorem B.1.

**Proof of Corollary B.2.** Finally we make a summary of this result for the proof of Corollary B.2. First we write the number of snapshot point calculation we need to achieve $\mathbb{E}_{\xi_{t-1}}F(x^t) - F^\star \leq \delta$ at the second stage:

$$N_s = \left\lceil \beta\sqrt{32 + \frac{12L}{n\mu_0}} \right\rceil \log_{\frac{\beta^2}{\alpha}} \frac{F(x^1) - F^\star}{\delta}. \tag{40}$$

When $\frac{2n\mu_c}{L} \leq \frac{3}{4}$, $N_s = O\left(\sqrt{\frac{L}{2n\mu_c}}\log\frac{F(x^1)-F^\star}{\delta}\right)$; when $\frac{2n\mu_c}{L} \geq \frac{3}{4}$, $N_s = O\left(\log\frac{F(x^1)-F^\star}{\delta}\right)$.

Hence it is enough to run $O\left((1 + \sqrt{\frac{L}{2n\mu_c}})\log\frac{F(x^1)-F^\star}{\delta}\right) \geq O\left(\max(1, \sqrt{\frac{L}{2n\mu_c}})\log\frac{F(x^1)-F^\star}{\delta}\right)$ epochs. Since we set the epoch length $m = 2n$ and hence the number of stochastic gradient $\nabla f_i(.)$ calculation is of $O(n)$. Therefore with some more trivial calculation we conclude that the complexity of the Rest-Katyusha algorithm is:

$$N \geq O\left(n + \sqrt{\frac{nL}{\mu_0}}\right) \log_{\frac{\beta^2\mu_c}{\mu_0}} \frac{\frac{1}{\rho(S_0+3)^2}\left[16(F(x^0) - F^\star) + \frac{6L}{n}\|x^0 - x^\star\|_2^2\right]}{\delta} + O(n)S_0. \tag{41}$$

$\square$

## C   Numerical test for different choices of $\beta$

In this section we provide additional experimental result on different choices of $\beta$. We choose to use the REGED dataset in this experiment as a example.

Figure 1: Comparison of different choices of $\beta$

**(C)**$\lambda = 2 \times 10^{-5}$ vanilla Rest-Katyusha      **(C)**$\lambda = 2 \times 10^{-5}$ Adaptive Rest-Katyusha

We test the Rest-Katsuha and Adaptive Rest-Katyusha on regularization level $\lambda = 2 \times 10^{-5}$ with 4 different choices of $\beta$ including the theoretically optimal choice which is approximately 2.7. However we found out that the choice of $\beta$ which provides the best practical performance is often slightly

Figure 2: Lasso regression on News20 dataset

Figure 3: Lasso regression on Sector dataset

larger in experiments for real datasets. For this specific example, we can see that the best choice for $\beta$ is 5 or 10 for both Rest-Katyusha and Adaptive Rest-Katyusha.

# D   Some more additional results

We provide here an additional large-scale sparse regression result on the benchmark *News20* dataset (class 1, the version by J. Rennie. *"Improving Multi-class Text Classification with Naive Bayes". 2001*) which sized 15935 by 62061, as well as the *Sector* dataset which sized 6412 by 55197, both of these datasets are available online on LIBSVM website. We also plot the $\ell_2$ distance towards a solution $x^\star$ where after a large number of iterations both of the algorithms will actually converge to. We can clearly see that for this specific case, minimizing the objective in a low precision is not enough to ensure that we are close to the solution, i.e. a $10^{-5}$ objective gap accuracy means only $10^{-1}$ accuracy on the optimization variable (geometrically the objective can be very flat along some directions).

# References

[1]  Z. Allen-Zhu. Katyusha: The first direct acceleration of stochastic gradient methods. *arXiv preprint arXiv:1603.05953*, 2016.