[Reviews · NeurIPS 2018]

Reviewer 1



# Summary This paper proposes a new method named Rest-Katsyusha which combines the noble method Katsyusha and restarting methods to accelerate convergence and make the method adaptive respect to the restricted strong convexity (although it is just a heuristics). By using the effective restricted strong convexity, this paper shows the accelerated linear convergence rates of Rest-Katsyusha up to a statistical tolerance. Moreover, the effectiveness of the proposed method is verified in experiments. # Detailed comments This paper is well written and easy to read. In short, motivation is also good. Since the strong convexity is unrealistic in high-dimensional settings, to relax it to restricted strong convexity is one of the major topic in the literature, recently. In addition, I think this paper is the first attempt to show an accelerated convergence rate up to a statistical tolerance in this setting. One concern is the readability of the proof of Theorem 3.4. * The notation of random variable (2) in supplement. line 16: \xi_k^t -> i_k^t ?. In addition, the number of stochastic gradient evaluations in each outer loop is $mS$ rather than $k$. Thus, indices in equation (2) is not good. * I think the proof is correct, but somewhat confusing because an inequality $\epsilon_t > \epsilon$ seems to be assumed without declaration.

Reviewer 2



I have read the response. The authors have addressed my question to satisfaction. ============================================== This paper presents an optimization algorithm for regularized ERM when the solution has low-dimensional structure (in the current paper, the specific structure is sparsity). The algorithm periodically applies Katyusha to the regularized ERM, while initializing each round of Katyusha from the approximate solution of previous round. When the strong convexity parameter is not known, the authors propose a line-search type of method for estimating it. The algorithm is demonstrated to have good performance on several datasets for solving the Lasso problem. While I could parse the results fine, I am not very familiar with the detailed techniques in this area, so my confidence is not high. My understanding is that Proposition 3.2 characterizes how close is the regularized ERM solution from the population optimum, thus optimizing the regularized ERM well means being close to the population solution. On the other hand, if one can establish the effective RSC for the regularized ERM, the objective becomes effectively strongly convex, even if the original objective is not globally strongly convex. It appears these two results were established by [3] and [36] previously. And the authors can then apply state-of-the-art algorithms for smooth and strongly-convex objectives, with Katyusha being a particular choice. From this perspective (if it is correct), the theoretical result presented here may not be very technically challenging to obtain. On the other hand, the method for locating the strong convexity parameter can be a practical contribution. Another question: In fact, if the RSC parameter mu_c is known and fixed, do we gain anything from restarting Katyusha, than simply running it once?

Reviewer 3



This work proposes to accelerate Katyusha for non-strongly convex functions by periodically restart it. This paper is well written and the motivation is clear. My main concern is on its practical performance and experiment. Empirically, the Rest-Katyusha generally performs better than Katyusha in terms of number of gradient evaluations. However, as shown in Figure 1, they are generally indistinguishable when the objective gap greater than $10^{-6}$ or even $10^{-8}$. In other words, the proposed method is effective most when an extremely accurate solution is required. Do we really need to solve the target problem so accurately in machine learning? To better show the advantage of Rest-Katyusha, the authors are suggested to test it on more real-world datasets. Especially, for the Lasso regression problem, it is better to use more high-dimensional datasets that is more reasonable in practice. Currently, for the used three datasets, only RCV1 can be consider as a high-dimensional dataset.